# Reducing Yield Asymmetry between Tension and Compression by Fabricating ZK60/WE43 Bimetal Composites

**DOI:** 10.3390/ma13010249

**Published:** 2020-01-06

**Authors:** Kangning Zhao, Dexing Xu, Xiao Song, Yingzhong Ma, Hongxiang Li, Jishan Zhang, Daolun Chen

**Affiliations:** 1State Key Laboratory for Advanced Metals and Materials, University of Science and Technology Beijing, Beijing 100083, China; zhaoning132@gmail.com (K.Z.); xudexingstar@gmail.com (D.X.); songxiaoustb@163.com (X.S.); myz001tx2@163.com (Y.M.); zhangjs@skl.ustb.edu.cn (J.Z.); 2Department of Mechanical and Industrial Engineering, Ryerson University, Toronto, ON M5B 2K3, Canada

**Keywords:** Mg/Mg bimetal composites, interface microstructure, texture evolution, yield asymmetry between tension and compression, deformation mechanism

## Abstract

In this study ZK60/WE43 bimetal composite rods were manufactured by a special method of hot diffusion and co-extrusion. Interface microstructure, deformation mechanism, and yield asymmetry between tension and compression for the composite rods were systematically investigated. It was observed that the salient deformation mechanism of the ZK60 constituent was {10-12}<−1011> extension twinning in compression and prismatic slip in tension, and different deformation modes resulted in yield asymmetry between tension and compression. In contrast, the WE43 constituent tends to be more isotropic due to grain refinement, texture weakening, solid-solution and precipitation strengthening, which were deformed via basal slip, prismatic slip, and {10-12}<−1011> extension twinning in both tension and compression. Surprisingly, it was found that yield asymmetry between tension and compression for the ZK60/WE43 composite rods along the extrusion direction was effectively reduced with a compression-to-tension ratio of ~0.9. The strongly bonded interface acting as a stress transfer medium for the ZK60 sleeve and WE43 core exhibited the coordinated deformation behavior. This finding provides an effective method to decrease the yield asymmetry between tension and compression in the extruded magnesium alloys.

## 1. Introduction

Currently, magnesium alloys are being applied as the structural materials in many important fields such as automotive, aeronautical, and space due to high specific strength and other outstanding properties [1,2,3,4,5,6,7]. However, to further widen the application of wrought magnesium alloy products, one of the key challenges to be solved is yield asymmetry between tension and compression after thermo-mechanical processing [8,9,10,11,12,13,14,15]. That is, for extruded magnesium alloy rods, the yield strength in tension (TYS) is largely different from the yield strength in compression (CYS) in the extrusion direction (ED) [3,8,11,12,13]. The origin of the yield tension-compression asymmetry is explained as follows: the presence of basal fiber texture causes {10-12}<−1011> extension twinning during compression along the extrusion direction, which is activated more easily than the tensile deformation in the highly extruded magnesium products. The polarity of {10-12}<−1011> extension twinning with a small critical resolved shear stress (CRSS) is one deformation mode activated more easily at room temperature. It indicates that, if {10-12}<−1011> twinning plays a crucial part during compression, the slip with a higher CRSS, such as prismatic <a> slip, can become the predominant mode of the Mg alloy rods during tension along the ED [16,17,18,19,20,21,22,23,24]. Thus, yield asymmetry between tension and compression will be produced.

Grain-size reduction, precipitation strengthening, and texture weakening are all proven to be effective in decreasing yield asymmetry between tension and compression [10,11,12,16,19,25,26,27,28,29,30,31,32,33,34,35,36,37]. For instance, Barnett et al. [16] found that the grain size affects yield stress and the sensitivity of twinning to grain size is obviously bigger than that of the slip. It was suggested that the yield tension-compression asymmetry for the coarse-grained magnesium alloys is 0.4–0.6, while it can decrease to 0.8–0.9 for fine-grained samples [9,11,12,28,37]. It was noticed that precipitates with a high density such as disk-like β’_2_ in the ZK series of Mg alloys can prohibit the propagation of {10-12}<−1011> twinning and harden extension twinning more than prismatic slip, leading to the dramatic suppression of yield asymmetry [10,29,30]. Rare-earth (RE) elements can weaken textures and suppress the extension twinning by solid solution strengthening in wrought magnesium alloys and as a result, the yield asymmetry is reduced [10,19,29,30,31,32,38]. For example, in WE43 and WE54 Mg alloys, after adding a range of RE elements, the plastic anisotropy is decreased, which is proven to be related to an increase of the CRSS value for twinning [19,29,30,31,32,39,40,41]. Some as-cast or RE-containing Mg alloys with a random texture also present a weaker yield asymmetry between tension and compression. The reasons could be imputed to the balance between the activity of the slip and the {10-12}<−1011> twinning induced by more random texture. It is further confirmed that yttrium is one of the most effective elements in reducing yield asymmetry due to texture weakening and precipitation strengthening [19,39,40]. Similarly, the ZK60 alloy also exhibits a reduced yield asymmetry between tension and compression due to the precipitation strengthening and the grain refinement caused by the Zr element [11,32,33]. Up to date, the investigations on the yield tension-compression asymmetry in the magnesium alloys mostly focused on the effect of grain size, pre-deformation, and alloying of single magnesium alloy materials. 

Recently it was reported that Mg/Mg bimetal composites were successfully prepared by several researchers, combining the advantages of different types of magnesium alloys and overcoming their disadvantages. As a result, the Mg/Mg bimetal composites are exhibiting an excellent application prospect instead of the traditional single magnesium alloys [42,43,44,45,46]. However, the corresponding study on the yield asymmetry between tension and compression for the Mg/Mg composite rods along the ED has not been reported. Therefore, in this study Mg-6Zn-Zr/Mg-4Y-2Nd-Gd (ZK60/WE43) composite rods with a well-bonded interface were first prepared by hot pressing diffusion and co-extrusion method, then the interfacial microstructure and mechanical behavior, especially the yield asymmetry of the ZK60/WE43 composite, ZK60 sleeve, and WE43 core along the ED, were investigated using the electron backscatter diffraction (EBSD) technique. The underlying mechanisms on the texture evolution during the deformation will be discussed in detail.

## 2. Experimental Procedures

### 2.1. Fabrication of ZK60/WE43 Composite Rods

The ZK60/WE43 composite rods were fabricated by using the homogenized ZK60 (sleeve) and WE43 (core) ingots represented in Figure 1. Firstly, the hollow cups can be machined by using ZK60 ingots. Then the WE43 bars were inserted into the ZK60 cups. To keep a robust bonding, the internal face of the ZK60 sleeve and the external one of the WE43 bars were polished, cleaned by alkaline, pickled by acid, and ultrasonically degreased by acetone. Then the composite ingots were heat treated in a muffle furnace at 723 K for 5 h by using a protective gas mixture of SO_2_ and N_2_. Afterwards, the composite ingots were extruded at 673 K by setting an extrusion rate of 0.6 mm/min and an extrusion ratio of 25. After extrusion, the composite bars were quenched in water and finally aging was performed at 475 K for 50 h. 

### 2.2. Microstructure and Mechanical Properties

Samples, 8 mm × 8 mm (radial direction × extrusion direction), were used to characterize the interfacial microstructure by setting the interface in the middle. Due to the contrast between the ZK60 sleeve and WE43 core, their bonding interface was easily distinguished. The microstructure of the specimens was observed by scanning electron microscope (SEM, ZEISS EVO-1, Carl Zeiss Microscopy GmbH, Jena, Germany) equipped with an energy dispersive X-ray spectroscopy (EDS). The detailed characterization of the precipitates was performed by a transmission electron microscope (TEM, Tecnai G2 F30, FEI Company, Hillsborough, OR, USA) by attaining high-angle annular dark-field scanning TEM (HAADF-STEM) images. To analyze the evolution of deformation texture and twins, different stages of compressive tests were carried out (i.e., the initial point (ε = 0%), compressive “yield” point (ε = 1.1%), and compressive “ultimate” point (ε = 9.4%)). For the electron backscatter diffraction (EBSD) characterization, the surface of the specimens was mechanically polished using Al_2_O_3_ suspension and electro-polished in AC2 electrolyte. EBSD characterization was performed at three locations composed of the interface, ZK60 sleeve, and WE43 core at the RD × ED surface to analyze the texture evolution. It should be noted that in the present round-bar co-extrusion, a radial direction (RD) of the composite rod is first selected, and the direction perpendicular to the selected RD is referred to as the tangent direction (TD). Pole figure (PF), grain boundary (GB), and inverse pole figure (IPF) from disparate zones were attained at post deformation: 86° ± 5° about <11-20> axis for extension twins, 56° ± 5° <11-20> for contraction twins, and 38° ± 5° about <11-20> axis for double twins [39,47].

The compressive and tensile tests of the ZK60 sleeve, WE43 core, and ZK60/WE43 composite rods were carried out in this study, respectively. For the compressive tests of ZK60/WE43 composite rods, bars with a diameter of 12.92 mm and a height of 19.38 mm (height/diameter ratio: 1.5) were selected. Similarly, the ones with a diameter of 3 mm and a height of 6 mm were used to characterize the compressive properties of the ZK60 sleeve and WE43 core. The details of samples cutting for the compressive tests are manifested in Figure 1. For the tensile tests, the composite samples with a gauge length of 19 mm, a gauge width of 2 mm, and a thickness of 1.4 mm were prepared by setting the interface in the one-third position of the gauge width. For comparison, the tensile samples for the ZK60 sleeve and WE43 core were kept at the same geometry and dimensions as the composites. The quasi-static uniaxial compressive and tensile tests were accomplished in a universal material testing machine along the extrusion direction (ED). For each mechanical test, at least three specimens were repeated to characterize the compressive and tensile properties. To ensure accuracy, the load-displacement data were corrected by machine compliance from the load frame.

## 3. Results

### 3.1. Interface Microstructure

As presented in Figure 2, a backscattered electron (BSE) micrograph of the co-extruded ZK60/WE43 composite rods displayed excellent contact between the ZK60 sleeve and WE43 core, and no cracks or voids at the interface were seen due to a uniform pressure applied during extrusion. The structure from left to right is ZK60 sleeve, diffusion zone, and WE43 core, respectively. Figure 2b shows the line scanning of Zn, Y, and Zr elements in the diffusion layer at a higher magnification. The contents of Y and Zr increased gradually across the interface from ZK60 to WE43. Figure 2c,d exhibits the elemental maps of Mg, Y, Zn, and Zr in the selected area of Figure 2b. It is revealed that the concentration of Zn decreased gradually from ZK60 to WE43 at the interface zone while the concentration of Y and Zr increased so strongly that a clear boundary could be observed. In this study, the thickness of the diffusion layer was regarded as the spacing across the interface where the Y or Zr elements are changed, which was evaluated to be ~20 µm.

Figure 3a,b shows the STEM micrographs cross the diffusion zone of the ZK60/WE43 composite and the corresponding bright-field TEM micrographs. This indicates that the dynamic recrystallization occurred since there were quite a few grains with a low dislocation density mixed with other ones with a higher dislocation density from Figure 3b. Meanwhile, a homogeneous distribution of the rectangular-like particulate phase was observed in this area. These dispersed particles were identified as Zn- or Y-containing particles. Figure 3c,d shows the selected area electron diffraction (SAED) images of points 1 and 2 in the image, respectively. Point 1 was identified to be YMgZn_2_ phase with facecentered cubic (fcc) structure and point 2 was a Y-rich phase, which was also a typical fcc crystal structure. The presence of Zn and Y elements in the diffusion zone indicated the occurrence of interdiffusion phenomenon in the process of hot pressing diffusion and co-extrusion.

### 3.2. Yield Asymmetry between Tension and Compression

Figure 4a–c describes the typical true stress-strain curves for the ZK60 sleeve, WE43 core, and ZK60/WE43 composites along the extrusion direction, respectively. From the tensile stress-strain curves, it can be found that three types of samples showed obvious work-softening behavior. Moreover, all the compressive curves had a plateau (i.e., a characteristic reflecting {10-12}<−1011> twinning) which dominated deformation. The mechanical properties attained from the plots are presented in Figure 4d. The ZK60 sleeve and WE43 core showed the yield tension-compression asymmetry along the ED. The CYS (194 MPa) of the ZK60 sleeve was lower than its TYS (230 MPa). Nevertheless, the CYS of the WE43 core (309 MPa) was higher than its tensile yield strength (293 MPa). The ZK60/WE43 bimetal composite showed a high yield strength (270 MPa) with a plastic strain of 0.15 in tension. In compression, the composite also showed a strong CYS (239 MPa). The ZK60/WE43 composite demonstrated a low yield asymmetry with a compression-to-tension ratio of 0.9. This indicates that, for the ZK60/WE43 composite, the yield asymmetry along the ED was effectively reduced. It was further noticed that the strengths (CYS and TYS) of the composite sample were much higher than those of the ZK60 sleeve in the same direction. In other words, the strength of the composite can be improved by the hot pressing diffusion and co-extrusion method.

### 3.3. Evolution of Crystallographic Texture

EBSD is introduced to analyze texture evolution in compression at three levels of true strain: 0%, 1.1%, and 9.4% along the ED. The pole figures of the ZK60 sleeve, WE43 core, and interface region with the {0001} and {10-10} planes are shown in Figure 5, Figure 6 and Figure 7. As widely acknowledged [2,3,11,13,28], this orientation is easy in forming extension twinning along the ED under compression. For the ZK60 sleeve, all the c-axes of the grains were almost perpendicular to the ED, indicating that the basal planes of the grains were mostly parallel to the ED as illustrated in Figure 5a. The original basal texture also showed a slight spread around the RD and its maximum intensity can reach ~8 MRD (multiples of random distribution). At a strain value of 1.1% (Figure 5b), basal {0001} poles still maintained an intensity of ~7 MRD with a little change of the c-axes. With increasing strain level from ε = 1.1% to 9.4%, a significant change was observed, and all the c-axes revolved towards the compressive direction or ED and the maximum intensity can reach ~18 MRD.

A texture evolution process of the WE43 core is shown in Figure 6. It is clear that the pole figure represents a larger extension of the basal texture towards the RD instead of TD for the WE43 core (i.e., most grains were orientated more randomly (Figure 6a) in comparison with that of the ZK60 sleeve (Figure 5a)). The low value of ~5 MRD in this pole figure indicates that the WE43 core showed a weak texture which is different from the magnesium alloys without RE such as the extruded AZ31, AM30, and ZK60 [5,10,12,13,14,23,33,37,48]. It indicated that the addition of RE elements can induce the slip and result in a weaker texture. At a strain value of 1.1% (Figure 6b), basal {0001} poles still maintained a high intensity of ~8 MRD with one pole between ED and RD. With the increase of strain level from ε = 1.1% to 9.4%, most of c-axes rotated towards the compressive direction or ED. However, for some grains, their c-axes still remained in the initial direction due to the difficulty in forming twins in some grain orientation. Thus, a maximum intensity of ~9 MRD was attained, which was about half of that of the ZK60 sleeve (~18 MRD, Figure 5c).

For the interface zone of the composite samples, all the c-axes of the grains were almost initially perpendicular to the ED. The intensity of original basal texture reached ~15 MRD while the c-axes of the grains almost pointed to the RD (Figure 7a). At a strain value of 1.1% (Figure 7b), basal {0001} poles keep a high intensity of ~15 MRD. With the increase of strain level from ε = 1.1% to 9.4%, all the c-axes almost revolved towards the compressive direction or ED and the intensity reached ~13 MRD, which was in-between the intensity of the ZK60 sleeve (18 MRD, Figure 5c) and WE43 core (9 MRD, Figure 6c).

Inverse pole figure (IPF) maps are demonstrated in Figure 8, Figure 9 and Figure 10 for the ZK60 sleeve, WE43 core, and interface zone, respectively. The orientation of the local crystal lattice was exhibited via color in the IPF orientation maps in regard to the crystal reference frame (i.e., ED or compressive direction in this study), based on the color legend seen in these figures. From the IPF orientation maps of the initial undeformed alloy, an almost equiaxed and un-twinned grain can be seen with a dual grain size of ~1.2 µm for WE43 core and ~2.8 µm for ZK60, respectively, calculated by a linear intercept method. As extension twinning was not profuse in the yield state (1.1%), the IPF orientation maps showed little change in color, which was consistent with the pole figures shown in Figure 5b. The twined grains will be discussed in the coming section. The grain orientation in the ZK60 sleeve evolved dramatically, which is revealed by the color change of the IPF orientation map at a strain level of 9.4% (Figure 8c). The movement of extension twinning leads to the c-axis nearly parallel (with rotations of 86.3°) to the compressive direction. The extension twin zones exhibit red in the IPF orientation maps. Owing to the fast extension of these twins, it would sweep the entire grains with the increase of plastic strain.

Figure 9 shows that there is little change with the grain orientations in the WE43 core, because most grains with c-axes between the ED (the compression direction) and RD were initially distributed more randomly (Figure 6a). Basal slip instead of the other deformation modes played a predominant role in compression along the ED, at early phase of deformation in particular. The movement of extension twinning combined with some basal slips at later phase of deformation also involved the rotation of c-axes to the ED.

The grain orientations in the interface zone (Figure 10) displayed a combined situation of ZK60 and WE43 in that the ZK60 side of the interface showed a similar feature to the ZK60 sleeve and the WE43 side showed a similar feature to the WE43 core, but the difference in the grain sizes on both sides was evident. Therefore, during the compression of ZK60/WE43 rods along the ED, the twinning on the ZK60 side and slip on the WE43 side will simultaneously start. Accordingly, the compressive deformation behavior of ZK60/WE43 rods differ from the individual ZK60 sleeve and WE43 core. Hence the interface zone will be discussed in more detail in the upcoming section.

## 4. Discussion

### 4.1. Deformation Mechanisms

By developing a crystal plasticity model, Qiao et al. [48] investigated the slip and twinning behavior of ZK60A during tension. For the stress along the ED, the prismatic slip becomes dominant. To verify this, Schmid factors (SFs) of the ZK60 sleeve relevant to the loading axis along the ED were calculated and Figure 11a–c presents the corresponding results for basal slip, prismatic slip, and extension twinning, respectively. The average SF of the basal slip approaches to ~0.15 (Figure 11a), suggesting that the basal slip is not easy to be motivated. That is, only if the external force is large enough and the resolved shear stress (RSS) surpasses the CRSS, the basal slip can be activated. Before the tensile yielding of the ZK60 sleeve sample, no rotation in the orientation of grains occurred, and the texture did not change obviously. The SF of the prismatic slip was ~0.45, which approaches the highest possible value of 0.5, suggesting that the prismatic slip is relatively easy to be activated. The prismatic slip becomes dominant during tension along the ED. Therefore, the tension along the ED has a higher yield strength. Generally, for the extruded Mg alloy (c/a = 1.624), the c-axis is in a tensile stress state when a compressive stress state is applied in the direction of the extrusion axis. Thus, the extension twinning {10-12}<−1011> was easily activated, and the extension twinning mode is a primary deformation mechanism during compression along the extrusion axis.

Jahedi et al. [19] and Davis et al. [25] claimed that the addition of RE elements such as Y, Nd, and Gd increased the strength and ductility, and reduced the anisotropy and tension/compression asymmetry which differ from other typical magnesium alloys such as AZ31, AM30, and ZK60 [5,10,12,13,14,23,33,48]. RE additions decreased the texture intensity of the wrought Mg alloys and suppressed extension twinning via solution strengthening and precipitation strengthening. Then the yield asymmetry of WE43 core can be reduced from grain refinement, precipitation strengthening, and texture weakening. The grain refinement is more easily realized, but the effect in decreasing yield asymmetry is limited. The low asymmetry of WE43 alloy is mainly orientated from precipitation strengthening during aging and the second phase formed on prismatic planes which hinders twinning and basal slip of dislocations. It was also reported that RE solute concentration played an important part in increasing twinning resistance [10,29,30,33]. The fundamental mechanisms on the increase in strength and ductility are related to the decreasing difference in the activation stresses for basal, prismatic, and pyramidal slip modes, the lessened twinning activity, and the formation of moderately strong texture. As depicted in Figure 11d–f, the SF of the basal slip and the prismatic <a> slip is ~0.33 under stress along the ED. However, the SF of extension twinning {10-12}<−1011> was close to ~0.41. Therefore, prismatic slip and basal slip are considered to be dominant in tension along the ED, especially at the beginning stage of deformation. During the compression along the extrusion direction, the plastic strain is accompanied mainly by the extension twinning; then the combined deformation mechanism, including the basal slip and extension twinning, occurred due to the higher SF value from the basal slip.

From Figure 11g–i, the ZK60/WE43 composite contains bimodal structures and the SFs of the basal slip, prismatic slip, and extension twinning {10-12}<−1011> are ~0.2, ~0.41, and ~0.42, respectively. The primary deformation mechanisms of the ZK60/WE43 composite would be a combination of those for ZK60 and WE43. Unavoidably, due to the evolution in the microstructure and mechanical performance at the interface zone, the stress concentration is easily generated. This will be further explained in the next section.

### 4.2. Twinning Evolution of ZK60/WE43 Composites

Figure 12, Figure 13 and Figure 14 show {10-12}<−1011> twin boundaries as highlighted by the color red. These figures confirm that {10-12}<−1011> is the preferential extension twin in ZK60 which is similar to other Mg alloys. To explain the mechanism on the yield asymmetry between tension and compression, EBSD grain boundary model was used to examine the microstructure evolution from the initial state, yield state, and “ultimate” state of deformation for the ZK60 sleeve, WE43 core, and interface zone.

The grain boundary maps of the ZK60 sleeve with different compressive strain amounts along with three selected zones and the corresponding pole figures (PFs) are illustrated in Figure 12. The twins were not seen in the initial state (Figure 8a and Figure 12a), similarly to Figure 8. With increasing strain, {10-12}<−1011> extension twins began to form, grow, and coalesce. For the deformation twins in zone 1 of Figure 12b,d, a set of twins were produced from the grain boundary intersection and due to the mutual connection of multiple twins, twin chains or twin bands were formed with increasing strain. In zone 2 of Figure 12b,e, twin lamellae were formed due to the larger grain and the suitable deformation condition with respect to the compressive axis. To explain the mechanisms on the selection of specific twins, SF values were calculated and the active twin planes and their tracks were detected by HexaSchmid software [20]. As summarized in Figure 12d,e, the twinning first occurs in blue grains under compressive stress and then in green grains along the twinning plane and direction due to the higher SF values (0.45–0.49) of blue grains than those of the green ones (0.36–0.41). Further compression resulted in the amalgamating of pre-existing extension twins or the vanishing of twin boundaries. Even after the “ultimate” state, there were some untwined grains left mainly in the green grains in Figure 8c and Figure 12c.

Using the red color for the extension twin {10-12}<−1011>, the twin boundaries are depicted in Figure 13, and Figure 9 demonstrates the corresponding IPF orientation maps. The {10-12}<−1011> twinning is also the preferred extension twinning in the WE43 core attained from these images. There is no evidence of substantial contraction twinning found at yield strain or larger “ultimate” strain. Owing to the refined grains, solid-solution, and aging strengthening, the amount of extension twinning in the WE43 core is smaller than that in the ZK60 sleeve. It is known that the basal slip mode is activated easily among all kinds of slip systems in magnesium alloys. Referring to the SF value shown in Figure 11d–f, the deformation mechanism is mainly the basal slip and twinning at the yield state for the WE43 core. Thus, the restraint of extension twinning and the activation of basal slip can decrease the yield asymmetry between tension and compression.

The microstructure in Figure 14a along with the corresponding IPF orientation maps in Figure 10a reveals that the extruded interface zone contains almost homogeneous as well as equiaxed grain structures with a mean grain size of ~3.1 µm, and it presents a representative basal texture, along with the c-axes of most grains presumably perpendicular to the ED on the ZK60 side. A relatively weak texture occurs on the WE43 side due to the fine equiaxed grain structure (~1.1 µm). Figure 14b shows the twin growth behavior in the selected grains after being compressed to yield point. The selected grains are explored and highlighted in Figure 14b. It is seen that, even though the green and blue grains displayed different SFs (0.38 and 0.44, Figure 14d) along the twinning plane and direction, a set of twins were formed from grain boundary intersection and multiple twins could link mutually to form twin bands or chains with increasing strain in the ZK60 side. A few lenticular twins appeared, with a few peaks of ~86° observed in the misorientation outline presented in Figure 14f, corresponding to the number of twin boundaries intersected by line A-B in Figure 14b. Meanwhile, in the WE43 side, the fraction of extension twins is much lower than that in the ZK60 side, even lower than that in the WE43 core (Figure 13) in the same deformation state. However, it is noted that some narrow banded contraction twins were formed within the amalgamated extension twins in the ZK60 side (Figure 14c,e). To verify the formation of contraction twins, the misorientation across these twin boundaries was ascertained by the outline along the direction determined by the dotted line C-D in Figure 14c. The numbers of peaks in Figure 14g matched that of twin boundaries shown in Figure 14c and all the misorientation peaks were ~56°, verifying that the contraction twins are present. It is the result of the stress concentration occurring at the interface while maintaining a high bonding strength.

More details of the plastic deformation are illustrated in Figure 15, which originates from twinning and double twinning shown in Figure 14c. In the deformation process, twinning and double twinning occur in the following steps: twinning nucleation, twinning growth, entire grain twinning, and/or double grain twinning. All of them are related to the general glide-shuffle or shear-shuffle involving the slip of twinning dislocations. Initiated with an un-twinning area (i.e., matrix grain shown in Figure 15a), the grain begins to twin by the nucleation associated with the {10-12}<−1011> extrusion twinning only if the RSS of the Mg substrate is equal to CRSS for twin nucleation. Then the grain is divided into an un-twinning zone (matrix) and a twinning zone (twin), as shown in Figure 15b. Their crystallographic lattice, which is mirrored through the twin boundaries (TBs) equivalent to an 86.3° rotation to the c-axis in hexagonal close-packed (hcp) unit cell (or basal plane as demonstrated in the dotted lines in Figure 15). After twining nucleation, twin growth starts when the RSS of the Mg substrate surpasses the CRSS value (Figure 15c). Twinning growth occurs via the glide of twinning dislocations (TDs) of the substrate on the TB when the stress generated on TDs acts as the driving force at the TBs. Due to the twinning with polar property, the TDs can slip along the twinning direction. With increasing strain, the entire grain is twinned, along with the slip of TDs at the TB, where extension twins merge and twin boundaries disappear, as shown in Figure 15d. Double twinning, representing the activation of a twin within a twinned grain (Figure 15e), occurs after the grain becomes totally extension twinning. In this case, the c-axis is in compression stress condition while it is compressed along the ED. Accordingly, the contraction twinning {10-11}<10-1-2> is activated (Figure 15e), which represents {10-12}-{10-11} double twinning sequence.

## 5. Conclusions

In the present study ZK60/WE43 composite rods were fabricated by hot diffusion and co-extrusion, and the effect of composite structure on reducing yield asymmetry between tension and compression was investigated. The evolution of crystallographic texture in the ZK60/WE43 composite rods at room temperature was examined during quasi-static tension or compression. The main findings are summarized as follows:
(1)In the ZK60 sleeve, a relatively high intensity of the crystallographic texture was present with most c-axes of the grains perpendicular to the ED. The predominant deformation mechanism was {10-12}<−1011> extension twinning in compression, while prismatic slip became a dominant deformation mode in tension along the ED. The different deformation modes were observed to be responsible for the plastic anisotropy of the extruded ZK60 sleeve.(2)In the WE43 core, the presence of yttrium resulted in fine-grained structure, texture weakening, solution strengthening, and second phase strengthening. In tension along the ED, the non-basal (prismatic) slip of <a> dislocations became prevalent. In compression, the extension twinning {10-12}<−1011> and basal <a> slip were activated. WE43 was relatively isotropic at room temperature since the activation stress was related to the combination of multiple deformation modes.(3)The yield tension-compression asymmetry between tension and compression for the ZK60/WE43 composite along the ED was effectively reduced with a compression-to-tension ratio of ~0.9. The strengths (CYS and TYS) of the composite samples lay in-between those of the constituents in the same direction.(4)EBSD results indicated the nucleation and growth of extension twinning and the presence of double twinning in a host of grains. The primary extension twins were observed to be merged together, covering the whole zones of grains with increasing strain. In the interface zone, the contraction twinning {10-11}<10-1-2> was also activated within the extension twin at a higher strain to form {10-12}-{10-11} double twins.

## Figures and Tables

**Figure 1 materials-13-00249-f001:**
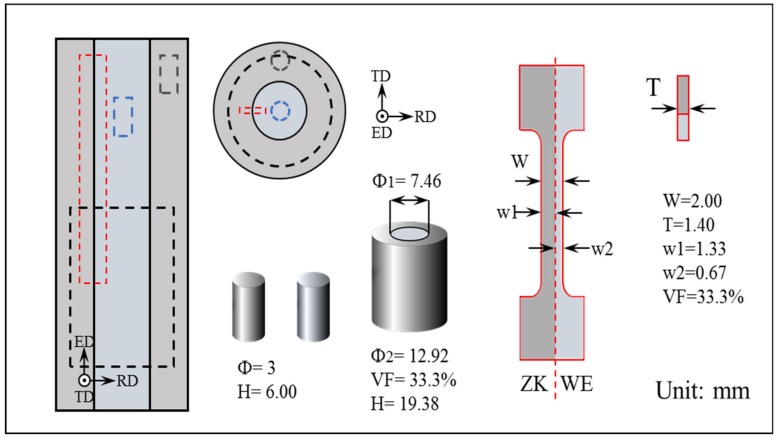
Schematic diagram showing the specimens for compressive and tensile tests.

**Figure 2 materials-13-00249-f002:**
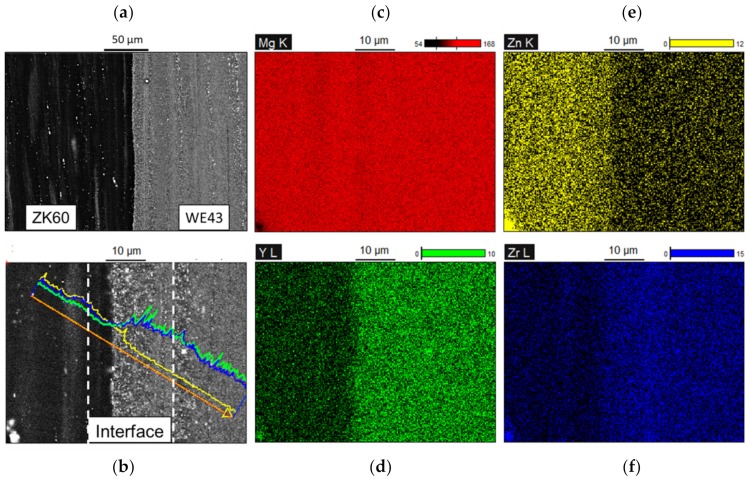
(**a**) Interfacial microstructure of the extruded ZK60/WE43 rods; (**b**) higher magnification micrographs and energy dispersive X-ray spectroscopy (EDS) line analysis; mappings of (**c**) Mg, (**d**) Y, (**e**) Zn, and (**f**) Zr elements.

**Figure 3 materials-13-00249-f003:**
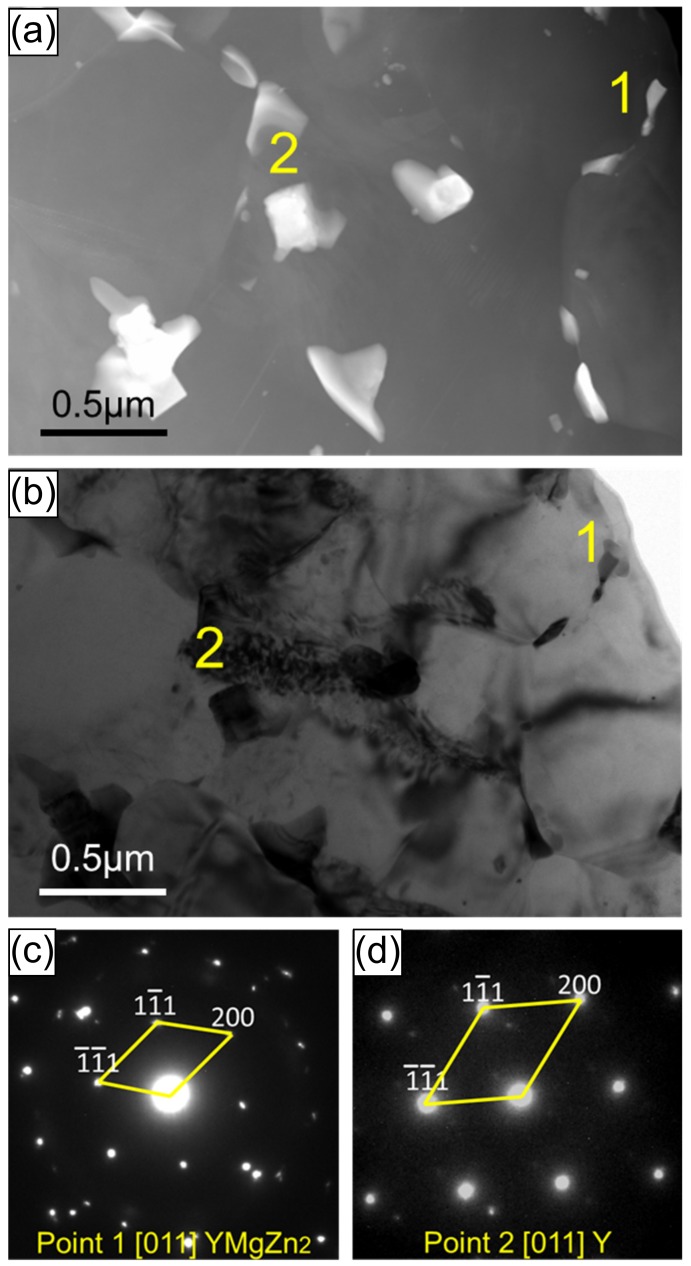
Scanning TEM (STEM) micrograph of the area across the diffusion region of the ZK60/WE43 composite. (**b**) Bright-field TEM micrograph of the same location, and the selected area electron diffraction (SAED) pattern for (**c**) point 1 and (**d**) point 2 indicated in (**a**) and (**b**), respectively.

**Figure 4 materials-13-00249-f004:**
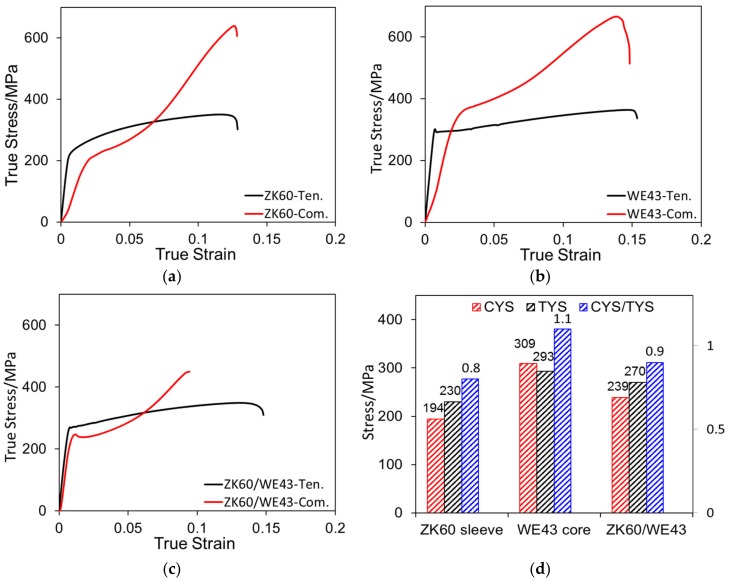
True tensile and compressive stress-strain curves of (**a**) ZK60 sleeve, (**b**) WE43 core, and (**c**) ZK60/WE43 bimetal composite along the extrusion direction. (**d**) Yield strength of the ZK60 sleeve, WE43 core, and ZK60/WE43 bimetal composite under compression and tension along the extrusion direction. Here Ten. and Com. represent the tension and compression, respectively, and CYS and TYS denote the compressive yield strength and the tensile yield strength, respectively.

**Figure 5 materials-13-00249-f005:**
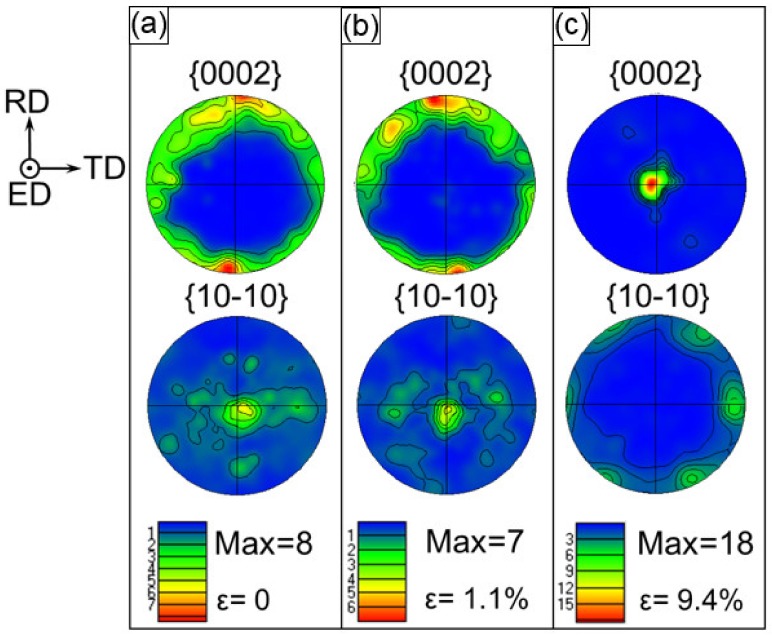
Stereographic pole figures illustrating texture of the ZK60 sleeve at different amounts of true strain at (**a**) 0%, (**b**) 1.1%, and (**c**) 9.4%, where the compression direction is parallel to the extrusion direction (ED).

**Figure 6 materials-13-00249-f006:**
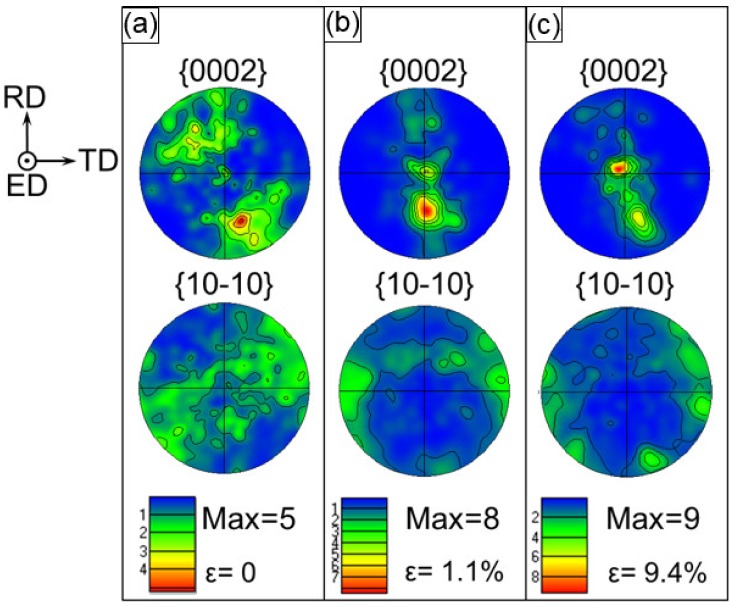
Stereographic pole figures illustrating texture of the WE43 core at different amounts of true strain at (**a**) 0%, (**b**) 1.1%, and (**c**) 9.4%, where the compression direction is parallel to the ED.

**Figure 7 materials-13-00249-f007:**
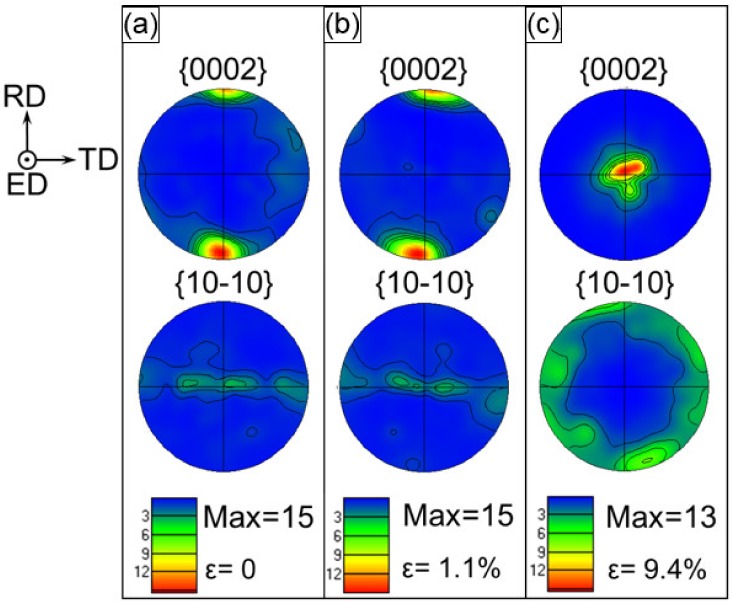
Stereographic pole figures illustrating the texture of the interface zone at different amounts of true strain at (**a**) 0%, (**b**) 1.1%, and (**c**) 9.4%, where the compression direction is parallel to the ED.

**Figure 8 materials-13-00249-f008:**
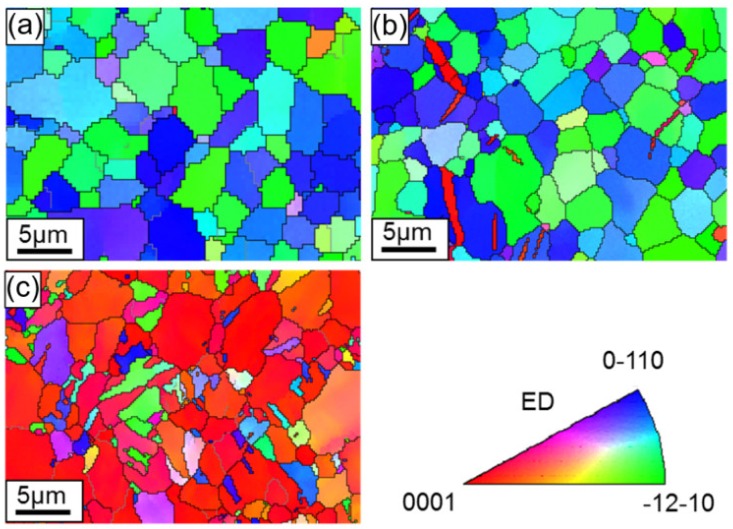
EBSD orientation maps, exhibiting microstructure in the ZK60 sleeve along the extrusion direction of the ZK60/WE43 composite rod, to different true strain levels: (**a**) initial undeformed state (0), (**b**) compressive yield point (1.1%) and (**c**) compressive “ultimate” point (9.4%). The compressive direction, i.e., ED, is perpendicular to the maps. The color in the maps represents the local crystal lattice frame with respect to the ED/compressive direction according to the color legend.

**Figure 9 materials-13-00249-f009:**
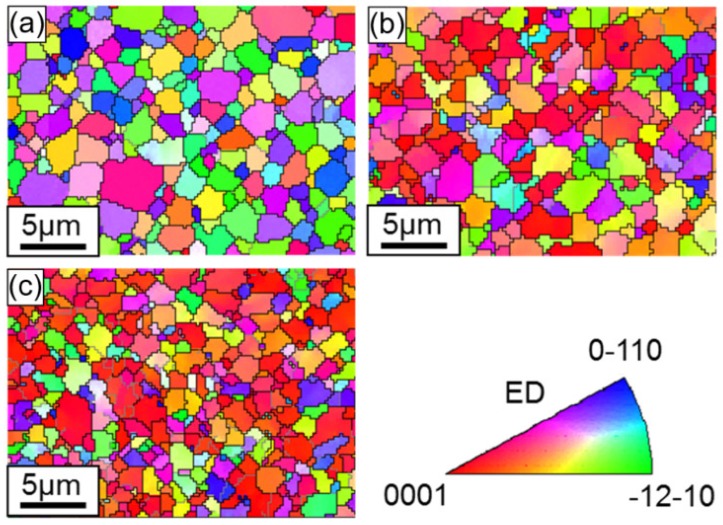
Electron backscatter diffraction (EBSD) orientation maps, showing microstructure in the WE43 core along the extrusion direction of the ZK60/WE43 composite rod to different true strain levels: (**a**) initial undeformed state (0%), (**b**) compressive yield point (1.1%), and (**c**) compressive “ultimate” point (9.4%). The compressive direction (i.e., ED) is perpendicular to the maps. The colors in the maps represent the local crystal lattice frame with respect to the ED/compressive direction according to the color legend.

**Figure 10 materials-13-00249-f010:**
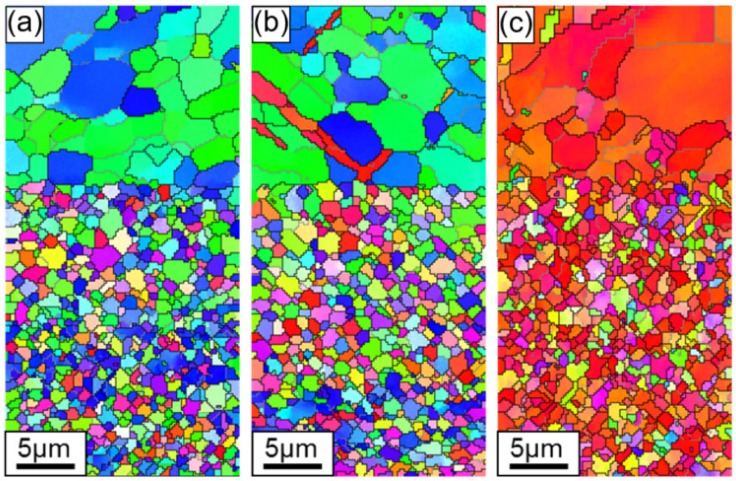
EBSD orientation maps showing microstructure in the interface zone along the extrusion direction of the ZK60/WE43 composite rod to different true strain levels: (**a**) initial undeformed state (0%), (**b**) compressive yield point (1.1%), and (**c**) compressive “ultimate” point (9.4%). The compressive direction (i.e., ED) is perpendicular to the maps. The colors in the maps represent the local crystal lattice frame with respect to the ED/compressive direction according to the color legend shown in Figure 8 and Figure 9.

**Figure 11 materials-13-00249-f011:**
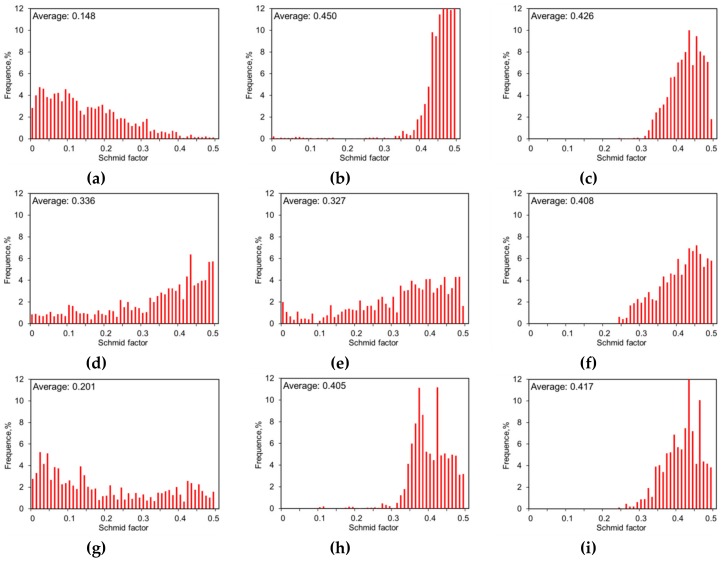
Schmid factors (SFs) versus relative spatial position in EBSD maps: basal slip in (**a**) ZK60 sleeve, (**d**) WE43 core, and (**g**) interface zone; prismatic slip in (**b**) ZK60 sleeve, (**e**) WE43 core, and (**h**) interface zone; extension twinning in (**c**) ZK60 sleeve, (**f**) WE43 core, and (**i**) interface zone.

**Figure 12 materials-13-00249-f012:**
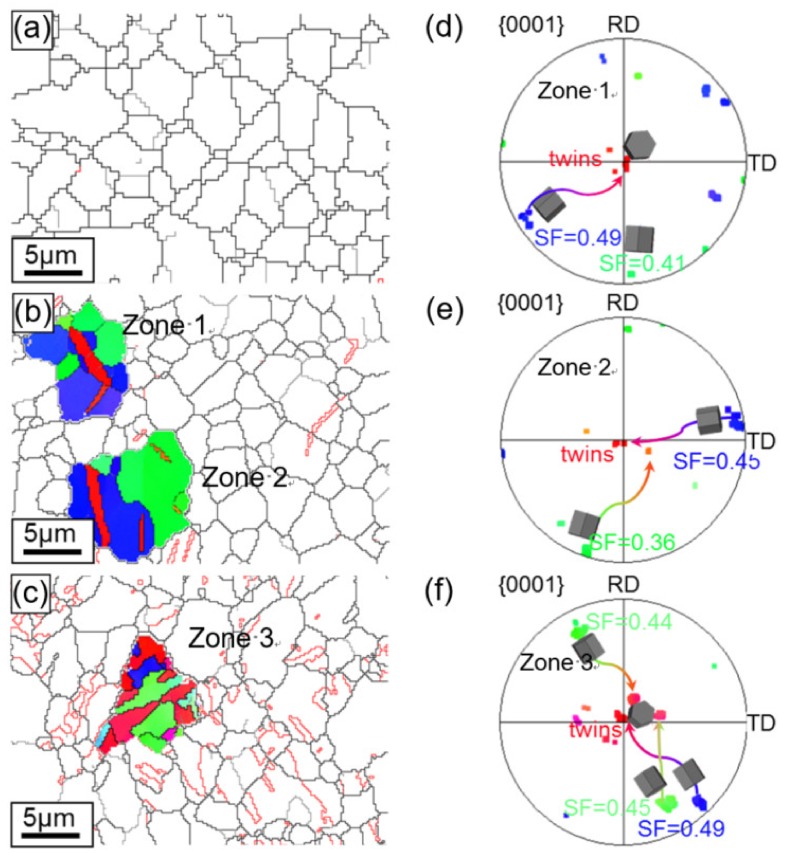
Twin boundaries in the ZK60 sleeve for the extension twinning {10-12}<−1011> mode recognized in the maps with a red color based on the expected twin-parent relationships. (**a**) Initial undeformed state (0%), (**b**) compressive yield point (1.1%), and (**c**) compressive “ultimate” point (9.4%). Crystallographic orientations of the selected grains are illustrated in (**d**) for zone 1, (**e**) for zone 2, and (**f**) for zone 3.

**Figure 13 materials-13-00249-f013:**
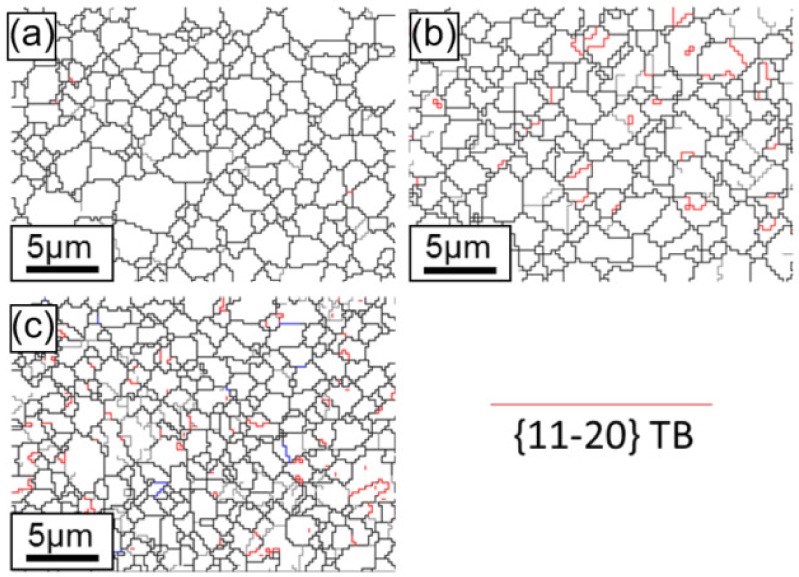
Twin boundaries in the WE43 core for the extension twinning {10-12}<−1011> mode distinguished in the maps with a red color as shown in the legend based upon the expected twin-parent relationships. (**a**) Initial undeformed state (0%), (**b**) compressive yield point (1.1%), and (**c**) compressive “ultimate” point (9.4%).

**Figure 14 materials-13-00249-f014:**
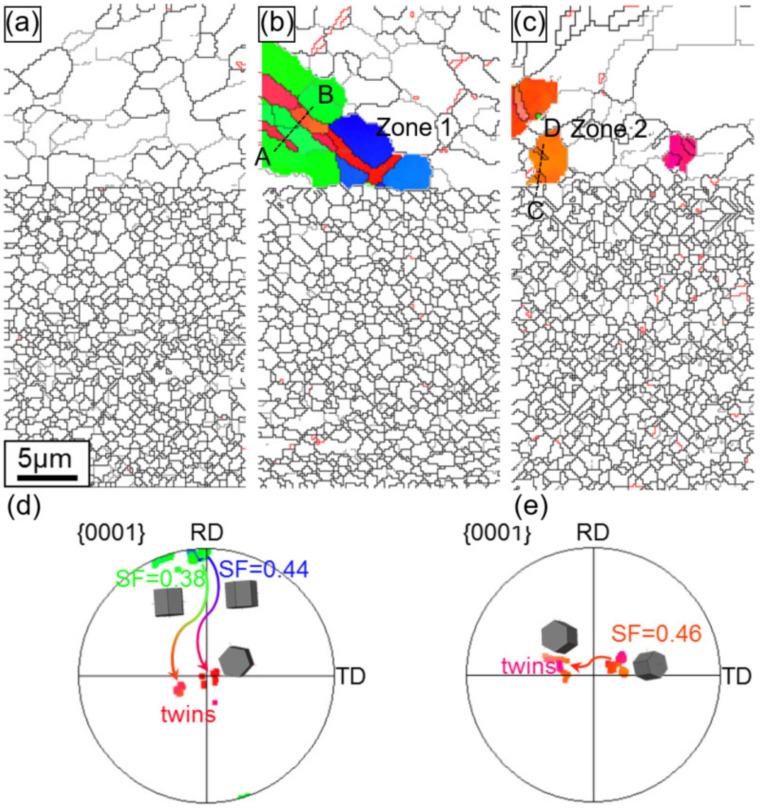
Twin boundaries in the ZK60/WE43 composite rod for extension twinning {10-12}<10-11> mode defined in the maps with a red color as specified based upon the expected twin-parent relationships. (**a**) initial undeformed state (0%), (**b**) compressive yield point (1.1%), and (**c**) compressive “ultimate” point (9.4%); (**d**,**e**) crystallographic orientations of the selected grains in (**b**) and (**c**); (**f**,**g**) misorientation profile along the direction determined by line A-B and C-D in (**b**) and (**c**).

**Figure 15 materials-13-00249-f015:**
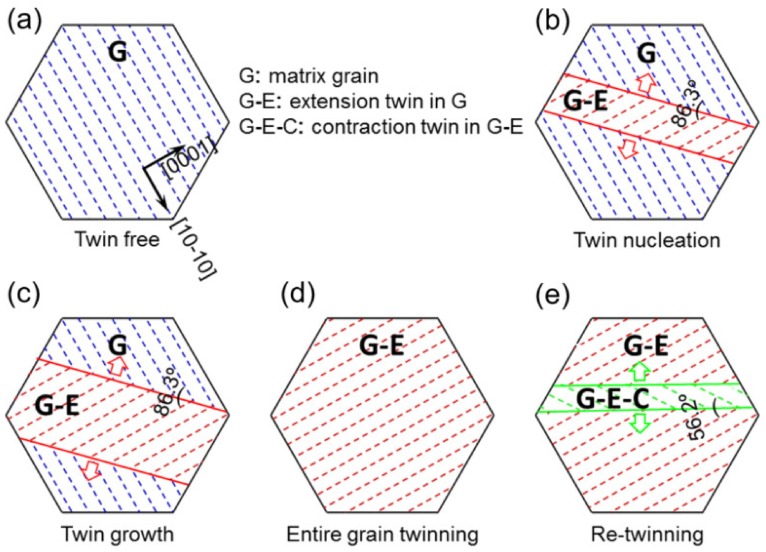
Schematic of twinning and double twinning in a grain. (**a**) The un-twinning grain (G: substrate grain), where the dotted blue lines stand for the matrix basal plane in a coordinate of [10-10] vs. [0001]. (**b**) Twinning formation: introduction of twin nucleation, where the solid red lines represent extension TBs, and the dotted red lines denote the twin basal plane (G-E: extension twin formed in G). (**c**) Twinning growth: growth of the extension twin with the nucleation and slip of twinning dislocations on the TBs. (**d**) Entire grain twinning: slip of twinning dislocations on the TBs to sweep the whole grain along with the vanishing of TBs. (**e**) Double twinning: introduction of a contraction twin (double twinning) in the twinned grain (G-E-C: contraction twin formed in G-E).

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
