# Peer review of "Reducing Yield Asymmetry between Tension and Compression by Fabricating ZK60/WE43 Bimetal Composites"

_materials, 2020, doi:10.3390/ma13010249_

Round 1

Reviewer 1 Report

The paper is well prepared, reporting on a mechanical behavior of a coaxial macro-composite consisting of two magnesium alloys, one acting as a core and one as a shell (called sleeve in the manuscript). The authors show that such a composite exhibits lower compression-tension asymmetry in plasticity than the individual materials.

I have just minor comments to the manuscript:

1) using the term re-twinning is a bit puzzling. One could either call it reverse twinning, or double twinning, but “re-twinning” somehow suggest a change to a new structure (similarly to re-crystallization, twin re-orientation, etc.). Also, as the manuscript discusses RE-alloyed magnesium, the reader can easily confused by using RE- and Re- with very different meanings in the same text.

Also, the term “twin within twin” (used on line 409) is confusing, as this term is often used for higher-order lamination in twinned structures, i.e., it means finer twins within coarse twins.

2) it is not very clean what does the term “Average” mean for Schmidt factors in Figure 11. Is that a weighted average over all possible Schmidt factors? If yes, then I cannot see the physical meaning for this. Using a median would be more meaningful, as the slips with very low SF do anyhow affect the slip initiations for those with high SF.

3) In the Conclusions, the authors claim that “the strengths (CYS and TYS) of the composite samples were also improved, which was higher than that of the ZK60 sleeve in the same direction”, which is hard to understand. The composite has higher CYS and TYS than the sleeve, that is right, but much lower CYS and TYS than the core. Saying that the performance of a composite which is somewhere between the performance of the constituents means “improving” the performance of one of them is strange – should the authors then say that the strengths were deteriorated, because they are lower than for the WE43 core?

Reviewer 2 Report

Overall this is an excellent research presentation.

Rarely do I find a paper that is comprehensive and yet written so well.  The figures are excellent.

If I had to find a flaw it is only to comment on words like lightweight which should be qualified.  However it is a very minor point.

The references are adequate - however aspects of dynamic-recrystallization and strain softening could have been addressed a bit better. Figure 4c could merit a bit more discussion particularly where the compressive true stress-true strain for compression shows much lower true strain prior to fracture when compared to the tensile curve.

Why do rare earths play such a significant out-sized role?

Finally conclusion 4 can be reworded as the first sentence makes no particular sense as the beginning of the fourth conclusion.

I am not sue that the EBSD shows nucleation of a twin.  Please consider whether you wish to assert this particularly as a conclusion.

Overall this is good to publish with very minor changes if at all.  Good job.
